

# Ozone profiles above Kiruna from two ground-based radiometers

Niall J. Ryan[1], Kaley A. Walker[1], Uwe Raffalski[2], Rigel Kivi[3], Jochen Gross[4], Gloria L. Manney[5]

[1]Department of Physics, University of Toronto, 60 St. George Street, Toronto, Ontario, M5S 1A7, Canada
[2]Swedish Institute of Space Physics, Box 812, SE-981 28 Kiruna, Sweden
5 [3]Finnish Meteorological Institute, Arctic Research Centre, Tähteläntie 62, FI-99600 Sodankylä, Finland
[4]Karlsruhe Institute of Technology, P.O. Box 3640, 76021 Karlsruhe, Germany
[5]NorthWest Research Associates, Socorro, New Mexico, USA and Department of Physics, New Mexico Institute of Mining and Technology, Socorro, New Mexico 87801, USA

*Correspondence to*: Kaley A. Walker (kaley.walker@utoronto.ca)

10 **Abstract.** This paper presents new atmospheric ozone concentration profiles retrieved from measurements made with two ground-based millimeter wave radiometers in Kiruna, Sweden. The instruments are the Kiruna Microwave Radiometer (KIMRA) and the Millimeter wave Radiometer 2 (MIRA 2). The ozone concentration profiles are retrieved using an optimal estimation inversion technique, and they cover an altitude range of ~16-56 km with an altitude resolution of, at best, 8 km. The KIMRA and MIRA 2 measurements are compared to each other, to measurements from balloon-borne ozonesonde 15 measurements at Sodankylä, Finland, and to measurements made by the Microwave Limb Sounder (MLS) aboard the Aura satellite. KIMRA is low-biased with respect to the ozonesonde data due to a general low bias in the KIMRA profiles around 22 km altitude, and MIRA 2 shows a smaller magnitude low bias and a high correlation coefficient. Both radiometers are in general agreement with MLS data, showing high correlation coefficients. An oscillatory bias with a peak of ± 1ppmv is present in the KIMRA ozone profiles over an altitude range of ~20-35 km, and is believed to be due to standing wave 20 features that are present in the spectra. A time series analysis of KIMRA ozone for winters 2008-2013 shows a local winter-time minimum in the ozone profile at approximately 35 km above Kiruna. The measurements are ongoing at Kiruna since 2002 and late 2012 for KIMRA and MIRA 2, respectively.

## 1 Introduction

Total column ozone ($O_3$) has decreased by approximately 2.5% over most of the planet during the 1980s and 1990s due to 25 increased emissions of chlorofluorocarbons (CFCs) (WMO, 2014). Thanks to the Montreal Protocol, and the ban on CFC emissions, total ozone columns have remained relatively unchanged since then, with recent indications of recovery (UNEP, 2015). Models suggest that these concentrations will recover to pre-1980 values by approximately 2060, but the projections are strongly dependant on future emissions of carbon dioxide ($CO_2$), nitrous oxide ($N_2O$), and methane ($CH_4$) (WMO, 2014). Long-term (seasonal, yearly, and decadal) measurements of stratospheric $O_3$ are an essential part of understanding how 30 compositional changes in the atmosphere are linked to the future radiative balance of the planet. Measurements of the Arctic atmosphere are particularly important as the Arctic is known to respond relatively rapidly to compositional changes that





affect the radiative budget of the planet (IPCC, 2013). With a likely upcoming gap in observations from profiling satellite instruments, ground-based instruments will represent the predominant source of atmospheric measurements needed to maintain a long-term $O_3$ profile record. Calibrations with respect to ground-based instruments are also needed in order to combine data from current and future satellites.

Millimeter wave radiometry is a well-established technique for atmospheric remote sensing. The technique offers the advantage of measuring radiation emitted from the atmosphere, allowing year-round measurements at high latitudes as there is no reliance on the sun as a source. High frequency resolution in measured spectra, and the relatively low Doppler broadening compared to pressure broadening in the atmosphere, allows the retrieval of altitude profiles of atmospheric composition with ground-based instruments.

Millimeter wave datasets of Arctic $O_3$ are sparse in time and in location. The Ozone Radiometer for Atmospheric Measurements (OZORAM) (Palm et al., 2010) has operated in Ny Ålesund, Spitsbergen (79° N, 12° E) since November 1994 and has been used for studies such as Palm et al. (2005), Langer et al. (1999). The Radiometer for Atmospheric Measurements At Summit (RAMAS) operated in Summit, Greenland (72° N, 38° W), briefly in 2003 (Golchert et al., 2005). The Kiruna Microwave Radiometer (KIMRA) has operated in Kiruna, Sweden since 2002 and the $O_3$ measurements have

been used to investigate of the 2002/2003 winter (Raffalski et al., 2005) and for validation of the satellite instruments: Global Ozone Monitoring by Occultation of Stars (GOMOS) (Meijer et al., 2004) and Michelson Interferometer for Passive Remote Sounding (MIPAS) (Steck et al., 2007) on board the Envisat satellite. The Millimeter wave Radiometer 2 (MIRA 2) has been housed at Kiruna on previous occasions and has been used to study the evolution of $O_3$ during the SOLVE/THESEO 2000 campaign (Kopp et al., 2003). Between 2004 and 2010, MIRA 2 was stationed at Pico Espejo,

Mérida, Venezuela (4800 m asl) and used for validation of the Sub-millimeter Radiometer (SMR) aboard the Odin satellite (Kopp et al., 2007). MIRA 2 has been installed indefinitely in Kiruna since November 2012.

The aim of this work is to develop and deploy operational inversion schemes that use the atmospheric spectra provided by KIMRA and MIRA 2 to retrieve $O_3$ concentrations above Kiruna, to assess the quality of these gas profiles through comparison with other $O_3$ measurements, and to examine the KIMRA data through an assessment of the wintertime

variability of $O_3$ above Kiruna. The inversion setup that was developed for this purpose can also be used for future measurements, as well as for older KIMRA data that have yet to be analysed.

Section 2 describes the instruments and datasets used in the study. Section 3 outlines the inversion setups used for KIMRA and MIRA 2. Section 4 shows the comparison of the retrieved $O_3$ profiles from KIMRA and MIRA 2. In Section 5, the KIMRA and MIRA 2 profiles are compared to measurements from ozonesonde instruments launched from Sodankylä, and

from the satellite-borne instrument, Microwave Limb Sounder (MLS) on Aura. Section 6 examines the KIMRA data by looking at the variability of wintertime $O_3$ concentrations above Kiruna from 2008 to 2013, and Section 7 offers some concluding remarks.





## 2 Ground-based instruments and datasets

### 2.1 KIMRA

KIMRA was partly designed by the Institute for Meteorology and Climate Research (IMK) at the Karlsruhe Institute of Technology (KIT) (Raffalski et al., 2002) and built at the Swedish Institute of Space Physics (IRF) in Kiruna, Sweden. The

instrument has been operated in IRF since 2002. KIMRA operates in the frequency range between 195 GHz and 233 GHz. The instrument has the capability to measure many species in this frequency range but, due to baseline issues, has only been used to measure $O_3$ and, since 2007, carbon monoxide (CO).

The detector in KIMRA is a Schottky diode mixer cooled to ~25 K within a cryostat. It has a single sideband (SSB) noise temperature of ~1800 K. The sideband filter is a Martin-Pupplett interferometer. A path length modulator (PLM) lies in the

beam path to supress standing waves. There are two blackbodies for calibration: at ~125 K and ~293 K for the cold and hot targets, respectively. The cold target is inside the cryostat. KIMRA has an acousto-optical spectrometer (AOS) with a practical bandwidth of 1.27 GHz and 1801 channels giving a resolution of ~0.7 MHz, as well as two Fast-Fourier-Transform spectrometers (FFTS). The narrowband FFTS is often centered on a nearby CO line and has been used in retrieving CO between 40 and 80 km (Hoffmann et al., 2011). The data from the AOS is presented here. A periscope-like mirror system,

with the sky mirror located in a dome on the roof of IRF, allows KIMRA to view in any direction on the sky. The elevation angle for each measurement is chosen automatically between 7° and 55° to give the highest signal-to-noise ratio (SNR) according to the tropospheric transmissivity. $O_3$ measurement durations range from 15 min to 360 min, depending on the atmospheric conditions. First technical descriptions of the instrument are given in Raffalski et al. (2002).

The KIMRA dataset presented here spans the time from 2008 to 2013, with some gaps in operation. The data used for

intercomparison of the retrieved $O_3$ profiles is from 20 November 2012 to 31 May 2013, and consist of 1152 retrieved profiles. The data used in examining the winter-time $O_3$ variability above Kiruna are from January to March over the years: 2008, 2009, 2010, 2011, and 2013. Data from January – May 2012, were not available. While measurement data from IRF exist, with interruptions, as far back as 2002, the KIMRA data used here from winter/spring 2012/2013 were selected to overlap with MIRA 2. The January – March data for the other years were selected because $O_3$ above Kiruna is expected to

have the most variation over this time due to chemistry and dynamics: this makes it an interesting dataset to study.

KIMRA looks only directly north or south for all of these measurements. The elevation angle changes for each measurement, which prevents the averaging of spectra, so each measured spectrum is inverted and any averaging of the retrieved profiles can be applied afterwards. As a result, the signal-to-noise ratio (SNR) changes for each spectrum due to atmospheric conditions and the measurement duration. The system's local oscillator (LO) is adjusted frequently (often every other

measurement) in KIMRA, which gives two differing spectral regions to be inverted. In one case, the LO is such that the CO line at 230.535 GHz is in the spectrum (KIMRA_O3CO measurement). In the other case, the centre frequency is shifted up by 34 MHz. This spectrum for this case is centred on the $O_3$ line and does not contain the CO line (KIMRA_O3O3 measurement). The reason for changing the LO was to see if incorporating more of the $O_3$ line in the spectrum would cause



the inversion to improve/differ (however, this result is not tested here). Slightly different inversion setups were needed to account for these different cases.

## 2.2 MIRA 2

MIRA 2 was developed at the Forschungszentrum Karlsruhe to measure $O_3$, ClO, $HNO_3$, and $N_2O$ between 268 and 280

5    GHz (Berg et al., 1998). The detector is a Shottky diode mixer cooled to ~25 K within a cryostat. The SSB noise temperature is ~800 K. The cold and hot targets are at ~47 K and ~300 K respectively. The cold target is located inside the cryostat. The sideband filter is a Martin-Pupplett interferometer. The sky mirror is contained within a removable periscope, which sticks out through a north-facing window. The MIRA 2 AOS has a bandwidth of 1.4 GHz, and 2048 channels giving a resolution of approximately 0.7 MHz. The spectra are centered on the $O_3$ line at 273.051 GHz. A PLM lies in the beam path to reduce the

effect of standing waves. A more detailed description of the instrument can be found in Berg et al. (1998).

The spectrometer is the same model as the KIMRA AOS and possible differences in the spectrometers are assumed to be negligible. The dataset from MIRA 2 presented here spans 1 December 2012 to 25 April 2013, and consists of 979 retrieved profiles. Inversions of measurements from May were not included because of instrumental problems. MIRA 2 continuously points north for all of the measurements presented here. As with KIMRA, the elevation angle of each measurement is

automatically chosen to give the best SNR according to the atmospheric conditions.

## 3. Inversion inputs and characteristics

### 3.1 Forward and inverse model parameters

The forward model used for these inversions is the second release of the Atmospheric Radiative Transfer Simulator: ARTS 2 (Eriksson et al., 2011). The inversion is done using the package Qpack 2 (Eriksson et al., 2005), which uses the Optimal

Estimation Method (OEM) (Rogers, 2000). Qpack 2 is designed specifically to work with ARTS, so one can perform forward modelling and retrieval work. Qpack 2 allows modelling of the instrument characteristics through sensor response matrices. The a priori volume mixing ratio (VMR) profiles used for the inversion of the KIMRA and MIRA 2 data are the Fast Atmospheric Signature Code (FASCOD) sub-Arctic winter scenario profiles (Anderson et al., 1986). The temperature and pressure information (zpTs) up to 50 km are from daily National Centers for Environmental Prediction (NCEP) profiles,

and above that is the U.S. Standard atmosphere. The forward model pressure grid is 200 layers that are evenly spaced in altitude between ground level and approximately 100 km. The retrieval pressure grid is 45 layers that are evenly spaced in altitude between ground level and approximately 90 km. The retrieved quantity is the fractional VMR, the VMR of the target gas as a fraction of the a priori for that gas. A polynomial of order three is included in the inversion and is fitted to each spectrum to account for some of the standing wave signal in the baseline. The inversions are nonlinear and a Marquardt-

Levenberg iterative root-finding method (Marquardt, 1963) is used with Qpack 2.





Attenuation of the signal due to the troposphere is accounted for by including the Millimeter wave Propagation Model MPM93 $H_2O$ continuum (Liebe et al., 1993) in the inversion. The spectroscopic parameters are taken from the HITRAN 2008 catalogue (Rothman et al., 2009). Estimates for the thermal measurement noise on each spectrum are obtained by fitting a second order polynomial to a relatively flat part of the spectrum (covering 400 channels), and calculating the

5 standard deviation of the residual for the fit. This value is used to calculate the measurement noise covariance matrix (Rogers, 1990). Error contributions from other instrumental and model parameters have previously been assessed for KIMRA and MIRA 2 using OEM and so they have not been repeated here. For KIMRA, the uncertainty in the retrieved profiles due to standing waves and systematic errors amounts to at least 1 ppmv (Kopp, 2001). For MIRA 2, an uncertainty of at least 1 ppmv is caused by errors due to standing waves, systematic errors, and thermal noise (Kopp, 2000).

**3.2 Example fits and properties of retrieved states**

Examples of fits to the data for a KIMRA_O3CO measurement and a MIRA 2 measurement are shown in Figure 1. There are substantial standing wave features in the spectra and the residua, often caused by internal reflections within an instrument. There are some relatively short-scale standing wave signals in both the KIMRA and MIRA 2 data that would require the inclusion of a large order polynomial in the spectrum. Polynomial orders up to nine were tried but no appreciable

difference was found in the results. Since some oscillations were apparent in the retrieved profiles, it was decided to increase the estimate of the noise on the spectrum by a fixed amount, different for each instrument. The offset was estimated based on the amplitude of the standing waves but the final value was found by adjusting the offset until oscillations were not clearly visible in the retrieved $O_3$ profiles. These oscillations are assumed to be due to the inversion fitting the short-scale standing wave features in the spectra, and while not clearly visible in a single profile, the results of the following sections show that

the effect of the standing waves is still present in the retrieved profiles.

The mean averaging kernels, measurement response, and altitude resolution for the retrieved profiles for each instrument are shown in Figure 2. The actual averaging kernels, and quantities derived from them, will vary for each measurement depending mainly on the SNR of the spectrum. The retrieval altitude range is chosen using altitudes that have a measurement response higher than 0.8. The choice of measurement response cut-off is somewhat arbitrary. 0.8 is used here as it limits the

25 contribution of a priori information in the retrieved profile and has been used for several similar ground-based instruments (e.g., Hoffmann et al., 2011; Straub et al., 2011). This cut-off gives a range of approximately 16 – 54 km. The altitude resolution is at best 8 km, and begins to degrade quickly above 40 km altitude. The values found here are very similar to previously shown values for these instruments (Kopp, 2001; Berg, 1998). The degrees of freedom for signal (DOFS) over the retrievable altitude range for the retrieved states are approximately 4 for each instrument.





## 4. Comparison of KIMRA and MIRA 2

### 4.1 Coincidence criteria

Since MIRA 2 always points north, only north-facing KIMRA measurements were used for this comparison. The measurement time and duration were used as follows to determine which profiles to compare to each other: For a given

KIMRA measurement, it was determined whether there are any MIRA 2 measurements whose midpoint in measurement time lies within the duration of KIMRA's measurement. If so, it was determined which measurement has a longer duration (say it was MIRA 2). Then it was checked whether there were any more KIMRA measurements that also had a midpoint that lay within the duration of the MIRA 2 measurement. If so, the KIMRA profiles from all of these measurements were averaged to produce a single profile that was considered coincident with the corresponding MIRA 2 profile. If not, the two

single profiles were considered coincident. This method compares profiles from measurements that overlap in time, and avoids using any measurement twice. 177 coincident sets were identified for the following comparison. The majority of the time differences between coincident measurements are less than 1 hour. Measurement durations for each of the instruments range from 15 minutes to 4 hours, with a mean time of approximately 1 hour for the coincident measurements.

### 4.2 Results of KIMRA and MIRA 2 Comparison

Figure 3 shows the comparison of coincident KIMRA and MIRA 2 $O_3$ profiles for December 2012 to April 2013. Both average VMR profiles in Figure 3(a) have lower values than the a priori VMR except for below about 18 km. The mean difference (KIMRA – MIRA 2) in the profiles shows an oscillatory structure that is largely within the limits of the plotted measurement error (summed measurement error from KIMRA and MIRA 2). The largest difference, which falls outside the measurement error range, is a negative value peaking at –1.1 ppm at 22.5 km. The standard deviation of the differences is

largest between 26 km and 34 km. There is a strong correlation of the VMR values (>0.95) above 35 km. This decreases to about 0.85 at 30 km and then drops rapidly to a minimum below 0.5 at 26 km. Within 17 – 24 km the correlation is above 0.70 before decreasing to ~0.50 at the lower retrieval limit.

Since there are approximately four DOFS for each measurement, the $O_3$ profiles were split into four altitude regions and the total $O_3$ concentration (in molecules per cm$^2$) was calculated for each region, corresponding to four $O_3$ partial columns. The

column densities were calculated using the temperature profiles from the zpTs and the ideal gas law. The altitude ranges of the four partial columns are: 16 – 26 km, 26 – 36 km, 36 – 46 km, and 46 – 56 km (the numbers are the centres of the retrieval grid layers) and each region corresponds to approximately one DOFS. The correlations of the $O_3$ partial column concentrations were calculated and a line of best fit was determined for each column. The fit was determined using a linear regression for data with errors in both the X and Y variables, following York et al. (2004). The results are plotted in Figure

4. The poorest correlation/fit is for the lowest partial column: the fit has a slope of 0.81 and an offset on the order of the average partial column concentration, and the correlation is 0.87. The best correlation/fit is for the 36 – 46 km partial column: the fit has a slope of 1.0 and an offset one magnitude lower than the average partial column concentration, and the





correlation is 0.97. The most variability in the differences is seen for the lowest altitude partial column. This corresponds to the altitude region where KIMRA has a relatively large low bias with respect to MIRA 2.

With just the data from these two instruments, it is difficult to diagnose the reason for the bias, however, the oscillatory structure in the mean difference of KIMRA and MIRA 2 profiles (Figure 3, middle panel) is present in all of the individual

difference profiles. This suggests a systematic error. It is reasonable to assume that this error is arising from the clear non-spectral line structures present in the spectra (see Figure 1). These structures are attributed to standing wave signals in the instruments. Standing wave signals in the baseline that have a scale similar to that of the spectral lineshape are the most difficult to separate from the sky signal, especially if the standing wave is symmetric about the line centre. For this reason, it is assumed that the standing wave signals in the KIMRA measurements will have more of an impact on the retrieved $O_3$ state

than the standing wave signals in the MIRA 2 spectra, which have a shorter scale in frequency space. Part of the variation in the differences between individual profiles would also be explained by standing waves as the cause, for the following reason: The retrieved ozone state is affected by the opacity of the atmosphere. If some of the standing wave in the spectrum is incorrectly attributed to a concentration of ozone at some altitude in the atmosphere, then that contribution to the retrieved state will also vary depending on the atmospheric opacity. A higher opacity will mean that the inversion attributes a greater

atmospheric concentration to the standing wave signal in the spectrum.

## 5 Comparison with ozonesondes and MLS

KIMRA and MIRA 2 $O_3$ profiles were compared to profiles from ozonesondes launched at Sodankylä, Finland (67.37°N, 26.63°E). The ozonesondes are launched by the Finnish Meteorological Institute at Sodankylä. The location of the launches is a good site for a comparison as it has a similar latitude to IRF (67.84° N) and the meridional gradient of $O_3$ tends to be

greater than the zonal gradient. Thirty-one ozonesonde measurements were provided for this study. The data are from between 31 October 2012 and 29 May 2013, and the sondes were launched approximately once per week. The instruments are electrochemical concentration cell (ECC) sondes, using a potassium iodide solution. The partial pressure of $O_3$ is calculated according to an electrical current produced by the reaction between $O_3$ and Iodide (Kivi et al., 2007; Smit et al., 2011). Overall uncertainty of the ozone measurements by ECC sondes in the stratosphere is about 5 % (Deshler et al., 2008;

Hassler et al., 2014). In Sodankylä the sonde preparation procedures have followed the generally accepted recommendations (Smit et al., 2011). The sounding system is DigiCORA III from Vaisala. The radiosondes are RS92-SGP (Dirksen et al., 2014). The radiosondes measure pressure, temperature, humidity and wind profiles during the balloon ascent and descent. Ozone data are transmitted using the Vaisala Digital Interface OIF92. The sondes used in this work have maximum measurement altitudes ranging from 18 km to 34 km, with an effective altitude resolution on the order of 100-150 meters.

The vertical resolution depends on the balloon ascent rate and the sensor response time. The ascent rate is typically 5 m/s and the response time of the ozone sensor is 20-30 seconds.



The Microwave Limb Sounder (MLS) is one of four instruments aboard the Aura satellite. The satellite is part of the National Aeronautics and Space Administration's (NASA) Earth Observing System. The MLS scans are synchronised to the orbit and measurements are at approximately the same time at the same latitude each day, spaced in distance by roughly 165 km on the suborbital track. The $O_3$ measurements used in this work are made using the spectral lines in the 240 GHz band.

More details on the instrument and observation technique are found in Waters et al. (2006). The v3.3/v3.4 version of the level 2 data was used in this comparison (Livesey et al., 2013). Gas concentrations are retrieved on a 55-layer pressure grid. The ozone profiles have a vertical resolution of ~3 km from 261 – 0.2 hPa, and 4 – 5.5 km from 0.1 to 0.02 hPa. These levels cover the "useful range" of the data quoted by the MLS team (Livesey et al., 2013). The precision of the measurements is ~0.04 ppmv from 215 – 46 hPa, 0.1 – 0.5 ppmv from 22 – 0.1 hPa, and 1.4 ppmv from 0.05 – 0.02 hPa. The validation of the

previous v2.2 data has been documented in Jiang et al. (2007), Froidevaux et al. (2008), and Livesey et al. (2008).

**5.1 Coincidence criteria**

Time, distance, and position relative to the polar vortex were the criteria used in determining which individual profiles to compare, and are described here for each instrument. The location of a measurement with respect to the polar vortex was determined using scaled potential vorticity (sPV) values (Manney et al., 2007). For the ground-based instruments the values

were calculated geometrically along the instrument's line of sight. An sPV value of $1.4 \times 10^{-4}$ s$^{-1}$, or nearby values, have been used extensively in previous works (e.g. Manney et al., 1994a, 2007, 2011; Jin et al., 2006) to define the vortex edge centre. Values of 1.6 and $1.2 \times 10^{-4}$ s$^{-1}$ have been used in the cited works to define the inner and outer edges, respectively, and the same values are used here. Both the north and south pointing measurements from KIMRA are used in the following comparisons.

The general procedure for ozonesondes vs. KIMRA/MIRA 2 and MLS vs. KIMRA/MIRA 2 comparisons was as follows: For a given ozonesonde/MLS measurement, a maximum limit on the difference in time between measurements was used to choose a group of possible KIMRA/MIRA 2 measurements with which to compare. This group was reduced to those measurements that were in the same location as the ozonesonde/MLS measurement, relative to the polar vortex (inside vortex/outside vortex/in vortex edge). From this new group, the closest measurement in space, using distance along a great

circle, was chosen as the measurement for comparison. Each KIMRA/MIRA 2 measurement was only used once in a comparison with another instrument.

For MLS, a maximum time difference of ±4 hours between measurements was used. A small time limit was preferred, and it was decided that less than 4 hours produced too few coincidences, while more than 4 hours did not make a significant difference to the number of coincidences. Either way, the choice of time criterion did not have a substantial effect on the

presented results (there was a slight increase in standard deviation). For the distance criterion, the closest measurement in space had to lie within a given latitude and longitude box: ±2° latitude and ±10° longitude. A smaller longitude box of ±5° was also tried but it halved the number of coincidences and made no significant difference to the results. The altitude at which the distance between measurements was calculated was 34 km. The reason for the choice is that this altitude coincides



approximately with the peak in the $O_3$ VMR profile above Kiruna, and it is also approximately the middle of the retrievable altitude range for the ground-based instruments. The altitude used for the sPV criterion for the MLS comparisons was 34 km, the same as that used for the distance criterion, explained above.

For the ozonesondes, no criterion was placed on the measurements with respect to their distance from KIMRA/MIRA 2

measurements. It is assumed that the location of the sonde $O_3$ profiles is above the launch station in Sodankylä. For the time criterion, the closest KIMRA/MIRA 2 measurement within 24 hours was used to compare to the ozonesonde profile, although the selected profile was almost always within a few hours of a ground-based measurement.  As the sonde profiles have differing maximum altitudes, the resulting comparisons have different numbers of coincident points for different altitudes. For the ozonesonde comparison, the criterion for sPV was applied at an altitude of 18 km, for the following

reasons: 18 km is the lowest maximum altitude of the used sondes, and it is the only altitude that is common to every sonde. Both the ozonesonde and MLS profiles were smoothed using the averaging kernels of the coincident KIMRA/MIRA 2 measurements before they were compared so that the profiles would have similar resolution. The effect of excluding $O_3$ measurements in the edge of the vortex was examined for each comparison and discussed in the following section.

**5.2 Results of comparison with ozonesondes**

Since the ozonesondes cover varying altitude ranges it was decided most appropriate to compare partial columns (calculated from 15 km to the maximum sonde altitude) with the ground-based data. Figure 5 shows the partial column densities from the ozonesondes plotted against the respective coincident measurements for KIMRA and MIRA 2. A linear regression was performed assuming that the ozonesonde data is true. The densities for KIMRA and MIRA 2 data were calculated as in Section 4.2 but the column heights were chosen to match the maximum heights of the coincident ozonesondes. The densities

for the ozonesonde data were calculated using the smoothed $O_3$ profiles and the temperature measurements made by the sondes during their flight. The KIMRA column densities are consistently less than those for the ozonesondes. The line of best fit to the scatter plot of the KIMRA and sonde data gives a relatively small slope of 0.44, and a correlation coefficient of 0.85. For MIRA 2, the slope of the line of best fit is 0.75 and there is a relatively high correlation coefficient of 0.98.

Figure 6 shows the sPV value for measurements made by each instrument at altitudes of 18, 24, and 30 km. The similarity of

the sPV for the ground-based instruments and the ozonesondes confirms Sodankylä as a good comparison site for Kiruna. Most likely due to the location of Kiruna and Sodankylä, the variability of sPV between altitudes shows that the measurements are sometimes simultaneously detecting $O_3$ concentrations from inside, outside, and within the edge of the polar vortex, depending on the respective altitudes. This makes the task more difficult when comparing measurements. Since the largest concentration gradients lie within the vortex edge, the results were examined to see whether it made a difference

if all $O_3$ profiles that lie in this region ($1.2 \times 10^{-4}$ s$^{-1}$ < sPV < $1.6 \times 10^{-4}$ s$^{-1}$) were excluded. The number of coincidences decreased from 25 to 17 for KIMRA. The partial column correlation increased from 0.85 to 0.86, and there was no change to the slope of the line of best fit. The change for MIRA 2 was more dramatic: the number of coincidences decreased from 14 to 7.  The partial column correlation increased from 0.98 to 0.99, and the slope of the line of best fit changed from 0.75 to



0.89. Although there are only 7 coincidences, these numbers indicate excellent agreement in the MIRA 2 and ozonesonde partial columns.

**5.3 Results of comparison with MLS**

Results of the profile comparison with MLS are shown in Figure 7 for KIMRA and in Figure 8 for MIRA 2 for the period of

December 2012 through April 2013. There are 359 coincident measurements for KIMRA and 361 coincident measurements with MIRA 2. Above 35 km, KIMRA has a good agreement with MLS with a consistent low bias of 0.3 ppmv. Below that, the oscillatory bias seen in the comparison with MIRA 2 (Figure 3, middle) is present, with max./min. of approximately +/- 1 ppmv. The steep drop in the correlation for KIMRA is also seen again (see Figure 3, right) at approximately 26 km. MIRA 2 shows better agreement with MLS. There is a general high bias with an oscillatory structure peaking at approximately

0.6 ppmv at the lowest retrievable altitude for MIRA 2. The bias ± the standard deviation of the differences is almost completely within the range of the sum of the measurement errors from MLS and MIRA 2. The correlation is above 0.90 for altitudes above 35 km and decreases to about 0.70 at the lowest retrieval altitude range of MIRA 2.

In the time series comparisons in Figure 9 and Figure 10 the profiles are split into four partial columns corresponding to the same altitude ranges as in Figure 3. The partial columns for MLS were calculated using the MLS temperature measurements

that are coincident with the MLS $O_3$ measurements. There is a clear increase in the column densities at the beginning of January for the KIMRA and MLS columns (although not as clear for the lowest column: 16-26 km). Lines of best fit were calculated accounting for errors in X and Y; again, a better correlation, fit, and variance are found for the MIRA 2 comparison than for the KIMRA comparison. The highest variance in the KIMRA measurements is seen in the lowest altitude partial column. Figure 11 shows the sPV values at 34 km for all KIMRA and MIRA 2 measurements, and all

coincident MLS data. One can see that most of the KIMRA and MIRA 2 measurements that lie within the edge of the polar vortex were removed from the comparison by the vortex criterion (see Section 5.1) because MLS was not also measuring inside the vortex at this altitude, and removing the remaining data that lay in the edge of the polar vortex made no significant difference to the results.

The $O_3$ profiles from all the KIMRA, MIRA 2, and MLS measurements from December 2012 through April 2013 were

averaged by day. The daily averages are plotted against time in Figure 12. Each dataset shows a similar evolution of $O_3$ over the winter: the generally low stratospheric $O_3$ concentration in December, a rapid increase in early January, and the descent of high $O_3$ concentration air, by about 10 km lower in altitude, through March and April. The differences in the profiles, found in Section 4.2, can also be seen: The low bias in KIMRA $O_3$ at ~22 km is apparent, and also the high bias at ~ 28 km, most easily noticeable in December and January. The higher MIRA 2 profile values around 40 km are attributed to a high

bias of ~ 0.5 ppm in the MIRA 2 data at this altitude: This conclusion is reached by comparing the differences in the KIMRA and MIRA 2 data (Figure 3) as well as their respective differences with MLS (Figure 7 and Figure 8).

It is important for the following section to see that the apparent "dip" in the $O_3$ profile, that one can see in January and February between approximately 28 and 38 km, is present in each dataset and is not a result of the oscillatory structure in the





KIMRA dataset. The high bias in KIMRA $O_3$ at ~28 km may make this dip more pronounced, but the low bias in KIMRA $O_3$ occurs at a lower altitude: ~ 22 km. It will be assumed from this point on that the KIMRA $O_3$ data has a low bias of 1ppm at ~22 km and a high bias of 1ppm at ~ 28 km, each maximum with a half-width of ~5 km.

## 6. The KIMRA dataset: daily variability in winter-time ozone above Kiruna

This section moves on from the comparisons between datasets into an examination of the KIMRA dataset over five years between 2009 and 2013, by looking at the daily variability of wintertime $O_3$ above Kiruna. Because one cannot definitively explain the variations in $O_3$ profiles at one location without using other data and/or model output, and that is not the aim of this work, only some general observations are made.

### 6.1. January to March, five-year $O_3$ time series

$O_3$ profiles have been retrieved from KIMRA measurements for January, February, and March, from 2008 to 2013. Data were not available from 2012. Daily averages were made and the resulting time series for each year are shown in Figure 13. January predominantly shows the lowest $O_3$ concentrations, except for in one region at around 30 km (particularly for 2009 and 2013), which has a sharp maximum. The sharp rise in $O_3$ concentrations in January 2009 and 2013 coincide with strong "major" sudden stratospheric warmings (SSWs) that started on 24 and 6 January in those years, respectively, and broke
down the vortex for about a month (e.g., Manney et al., 2015). There was a similar SSW in 2010 that began on 26 January and the KIMRA data show an increase in mid stratospheric $O_3$ concentrations a few days later. The increase in concentrations in late February 2008 also coincides with a brief SSW that occurred at that time. The most interesting feature is the aforementioned dip in the profile. The $O_3$ dip is present for some period of time in each year and disappears in late February or March. It is persistent up to the end of March in 2009. It is very unlikely that this feature is caused by chemical
ozone depletion as ozone loss resulting from heterogeneous reactions in the lower stratosphere has never been seen extending to this altitude in the Arctic (e.g., Manney et al., 2003, 2015; Kuttippurath et al., 2010; Livesey et al., 2015). A strong $O_3$ dip (most similar to 2010 presented here) has been observed previously with KIMRA in the winter of 2002/2003 (Raffalski et al., 2005), and coincided with ozone mini-holes between 4 and 11 December 2002, as reported by the European Ozone Research Coordinating Unit (EORCU). Mini-holes are attributed primarily to dynamical effects: an increase in the
tropopause height, coinciding with a cold lower stratosphere and transport of low-latitude air to high latitudes in the upper troposphere (Hood et al., 2001; EORCU, 2013). The persistence of the feature above Kiruna shown here is not consistent with an ozone mini-hole as the mini-holes typically last about a week before local recovery of the ozone and their locations are highly variable (e.g., Peters et al., 1995; Hood et al., 2001; Allen and Nakamura, 2002). The KIMRA measurements for winter 2002/2003 shown in Raffalski et al. (2005) still show the structure of the $O_3$ dip throughout most of December, with a
reduced magnitude compared to the instance of the ozone-mini holes. The latitudinal extent of the polar vortex has been shown to vary with altitude (e.g., Schoeberl et al., 1992; Manney et al., 1995; Harvey et al., 2002), which could explain an



occurrence of a local minimum/maximum, but such a feature would not remain stable long enough to account for the observations shown here. A possible explanation for the observed shape is the combination of downward motion of air within polar vortex, and transport of extra-vortex air into the middle to upper stratosphere, due to wave activity, for instance (e.g. Manney et al., 1994b, 1995, and 2015; Kuttippurath et al., 2012). The downward motion of vortex air would increase

$O_3$ concentrations in the lower stratosphere and the transport of extra-vortex air would increase $O_3$ concentrations in middle to upper stratosphere. This could give a local minimum in between, but the timing and extent of these processes can vary strongly interannually (see references in this section) and so further study is needed.

The monthly mean $O_3$ profile for each year is shown in Figure 14, as well as the overall monthly mean and standard deviation. For each month, $O_3$ shows the most variability in the middle stratosphere, between approximately 20 and 33 km.

Some of the variation may be due to the fact that KIMRA switches its observation direction between north and south, but the daily averaging of the profiles should remove much of it. Additionally, as discussed in Section 4.2, the standing wave signals in the $O_3$ spectra will cause systematic biases that can vary as a function of atmospheric opacity, and add to the natural variance in the profile. The peak in $O_3$ at about 35-40 km (can be seen in the a priori) is, on average, lowest in January and increases through March. The lower altitude peak at about 25 km decreases in March relative to January and February. This

general change in the altitude of the global maximum of $O_3$ can be expected if the polar vortex weakens and breaks up toward the end of winter, but individual years show strong variation in vortex break-up and reformation times related to SSWs and other variations in wave activity.

In the left panel of Figure 14, for January and February, every year shows an $O_3$ dip to some degree (most pronounced in 2010 and least in 2008), and at varying altitudes. These features tend to decrease in magnitude in March, except for 2009,

which maintains an $O_3$ dip greater than 1 ppm. While it is emphasised here that this feature is not a result of the described biases in KIMRA $O_3$, it will be difficult to quantitatively separate the two features.

## 7. Conclusion

The aim of this work is to develop and deploy an inversion scheme for the KIMRA and MIRA 2 instruments at IRF, Kiruna, and to characterize the retrieved $O_3$ profiles, through comparison with each other and with $O_3$ profiles from ozonesondes

launched from Sodankylä and from Aura MLS. KIMRA and MIRA 2 $O_3$ profiles from November 2012 to May 2013 were used in the comparison. The retrieval altitude range for the $O_3$ profiles is approximately 16 – 54 km, with a resolution of at best 8 km, derived from the full width at half maximum (FWHM) of the averaging kernels. KIMRA and MIRA 2 profiles show generally good agreement with each other. The mean of the difference in their profiles (bias) lies within the range of the measurement errors except for values between 18 and 24 km.

An oscillatory bias was identified in the KIMRA data, present in the comparison with all three instruments. There is a low bias of ∼ 1 ppm at 22 km, and a high bias of ∼ 1 ppm at 28 km, both with a halfwidth of ∼ 5 km. MIRA 2 shows a similar oscillatory bias, but with smaller amplitude (< 0.5 ppm) and finer altitude structure, that covers the whole retrieval altitude



range. The relatively stable oscillatory structures present in the profiles are assumed to be caused by standing wave signals that are clearly present in the spectra. Both KIMRA and MIRA 2 otherwise show generally good agreement with MLS $O_3$ profiles, and KIMRA shows a general low bias with respect to the ozonesondes. MIRA 2 shows overall better agreement with MLS and the ozonesondes, compared to KIMRA.

The multi-year KIMRA $O_3$ dataset was examined by using profiles from January to March, 2008, 2009, 2010, 2011, and 2013, to explore the local day-to-day variability of $O_3$ above Kiruna. The middle stratosphere between approximately 20 and 33 km shows the most variability. The lowest $O_3$ concentrations are found in January, and tend to increase through March. The location of the maximum in the $O_3$ profile shifts from ~30 km in January to ~39 km (the location of the maximum in the a priori profile) in March. The most interesting feature in the data is a local minimum in the $O_3$ profile, present to some

extent in all years and can be persistent for time scales larger than two months. The feature may be difficult to quantitatively assess with KIMRA because it tends to partially overlap in altitude with the oscillatory bias in the KIMRA data. Previous measurements with KIMRA during winter 2002/2003 showed a similar local minimum in the $O_3$ profile throughout most of December 2002.

### Acknowledgements

This work was supported by the NSERC CREATE Training Program in Arctic Atmospheric Science, the Centre for Global Change Science of the University of Toronto, the Institute of Space Physics, Kiruna, and the German Federal Ministry of Education and Research. Thank you to William Daffer at Jet Propulsion Laboratory, California Institute of Technology, for help in providing the derived meteorological data used in this work. Thank you to Philip Heron for the geographical plots.

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



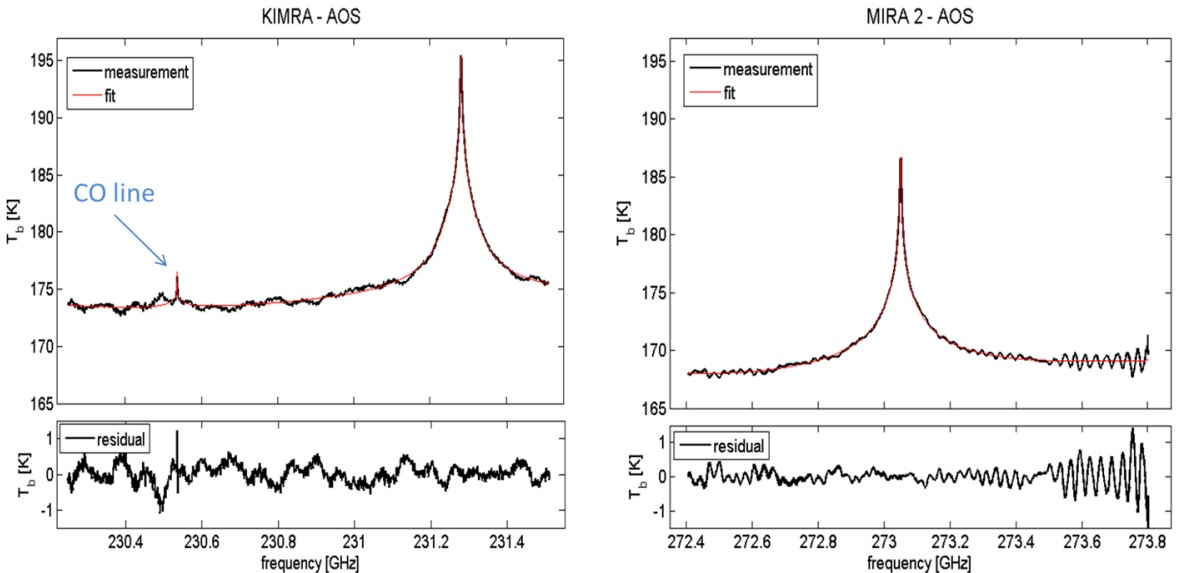

**Figure 1: Example spectra and fits for a KIMRA_O3CO measurement and a MIRA 2 measurement. Note: the visible oscillatory pattern in the baseline of MIRA 2 spectra has recently been eliminated after servicing of the AOS spectrometer.**

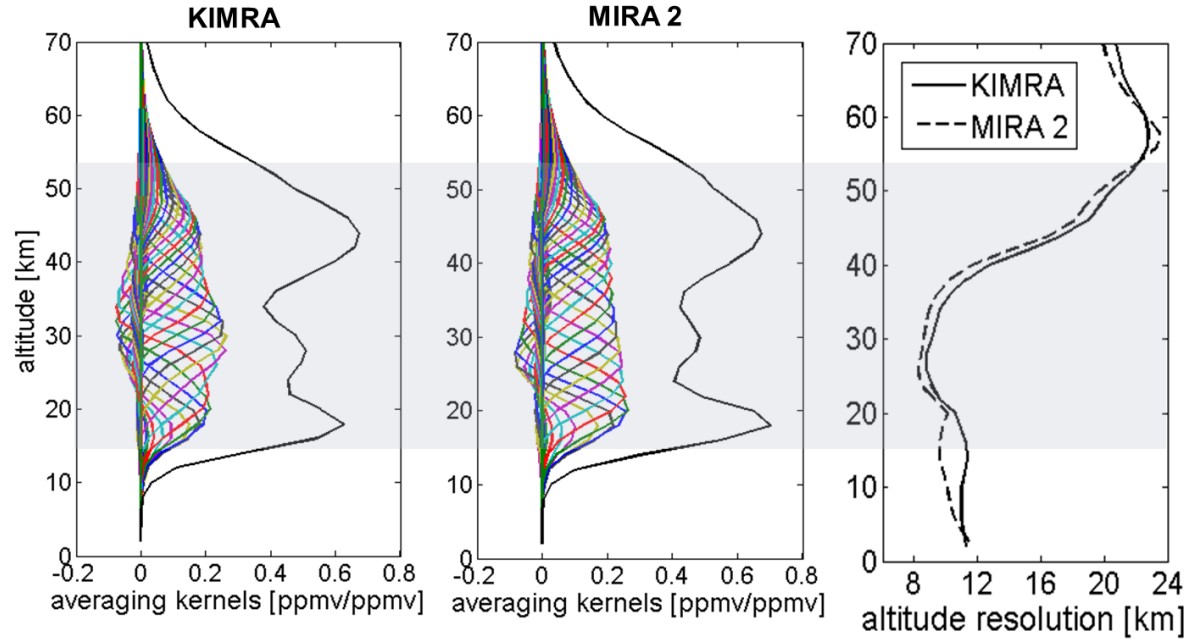

**Figure 2: Left and middle: mean averaging kernels for KIMRA and MIRA 2 O$_3$ retrieved profiles used for the comparison. The measurement response divided by 2 is also shown as the solid black line. Right: the altitude resolution of the profiles, given by the full width at half maximum (FWHM) of the averaging kernels. The shaded area represents altitudes with a measurement response greater than 0.8.**





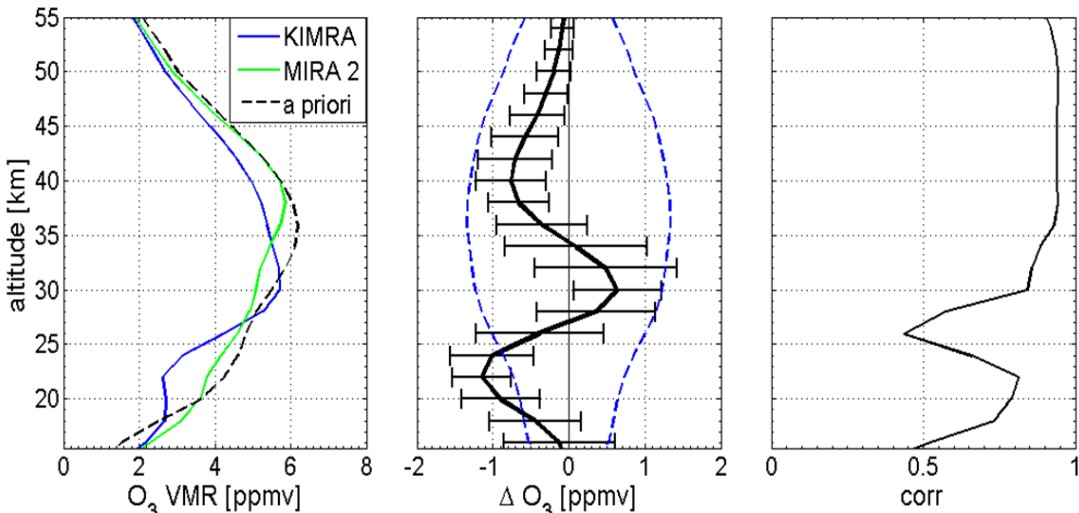

**Figure 3:** Left: the average of the 177 coincident $O_3$ profiles for KIMRA and MIRA 2 and the a priori for the inversions. Middle: the mean of the difference (KIMRA – MIRA 2) for coincident profiles (black) with ± the standard deviation as the error bar. Also shown is the sum of the average KIMRA and MIRA 2 measurement error (dashed blue). Right: the correlation of the coincident pairs at each altitude.

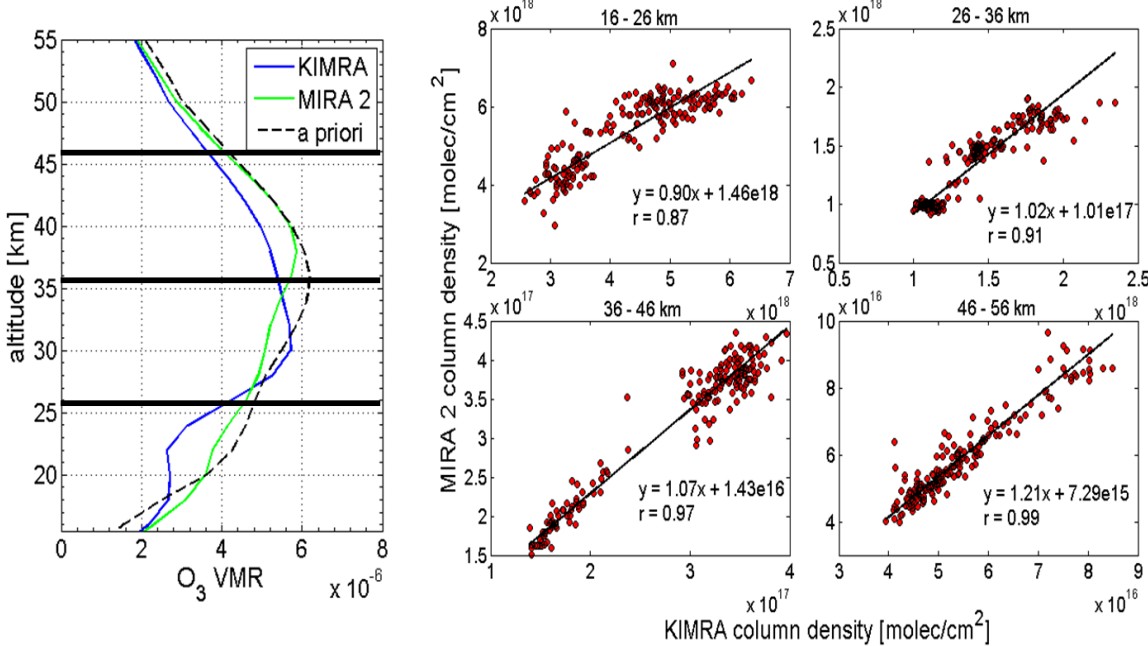

**Figure 4:** Left: same as left plot in Figure 3 (in ppv) but with markers showing the altitude separation of the partial columns. Right: scatter plot of the partial columns of coincident KIMRA and MIRA 2 data, including a line of best fit. The correlation for each of the sets of partial columns is also shown.




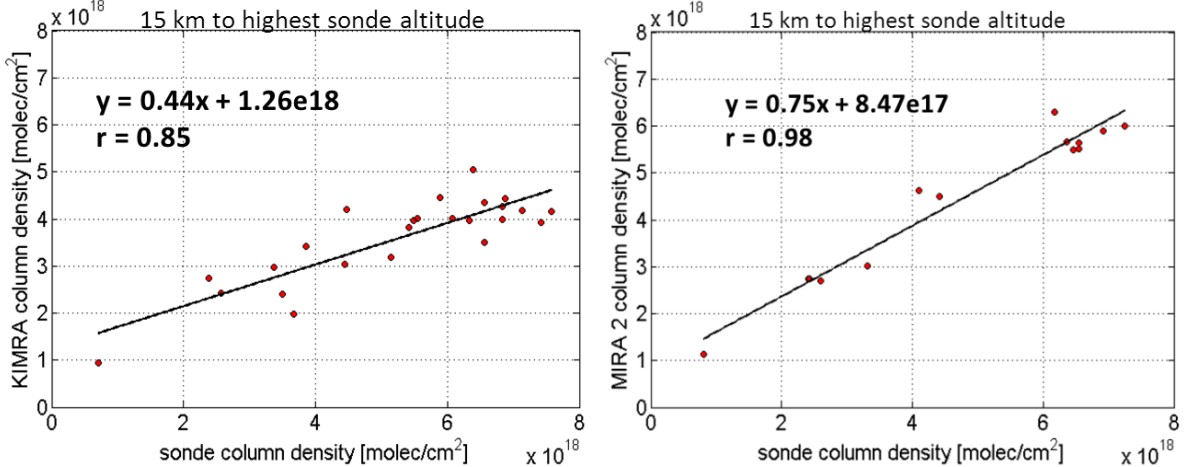

**Figure 5: Left: scatter plot of the KIMRA O₃ partial columns against Sodankylä ozonesonde partial columns from November 2012 through May 2013. The line of best fit is shown with the accompanying equation and correlation coefficient. The data has not been filtered by position relative to the polar vortex. Right: The same as left plot but for MIRA 2 and Sodankylä ozonesonde measurements from December 2012 through April 2013.**

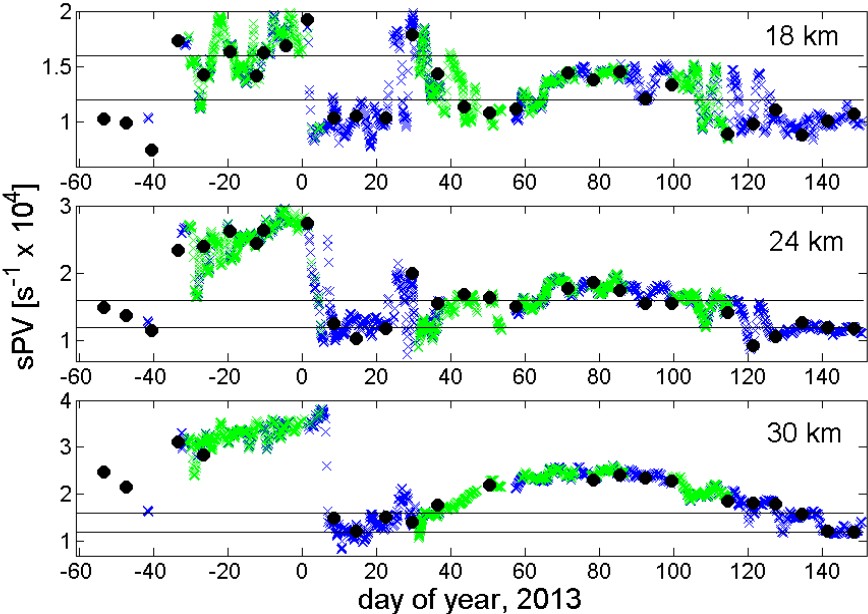

**Figure 6: The sPV value corresponding to the measurement locations of KIMRA (blue X), MIRA 2 (green X), and Sodankylä ozonesondes (black dot), at altitudes of 18, 24, and 30 km. Lines of sPV values of 1.2 and 1.6 x 10⁻⁴ s⁻¹, respectively defining the outer and inner edge of the polar vortex, are also shown.**



**Figure 7: Upper row: the same as for Figure 3, but for KIMRA – MLS. Lower left: histogram of the time difference between coincident measurements. Lower right: map showing the locations of the coincident MLS measurements (magenta circles), Kiruna (yellow triangle), and Sodankylä (cyan triangle).**





**Figure 8: The same as Figure 7 but for MIRA 2 – MLS.**



**Figure 9: Upper: the O$_3$ partial column densities from KIMRA (blue-filled squares) and MLS (hollow magenta diamonds) measurements plotted against time. Lower: scatter plots of the corresponding coincident data. Lines of best fit and correlation coefficients are shown.**





**Figure 10:** Same as Figure 10 but for MIRA 2 (green-filled squares) and MLS (hollow magenta diamonds).



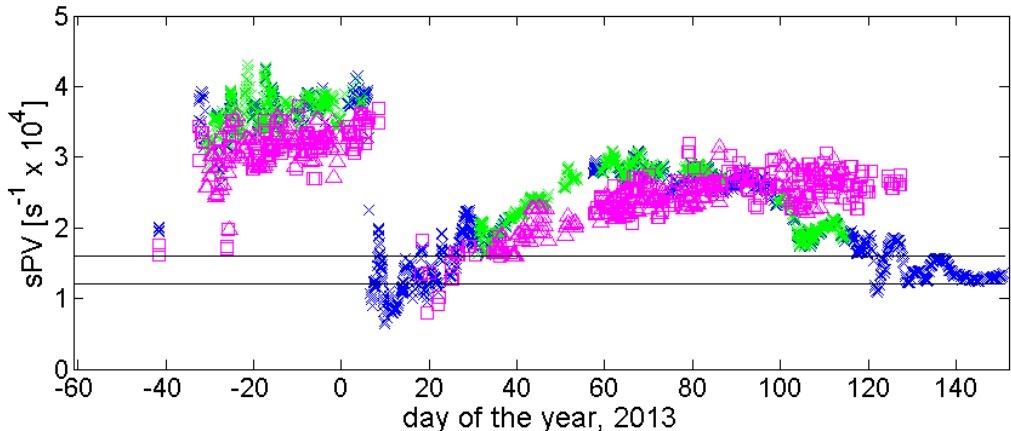

**Figure 11: sPV values at 34 km at the location and time of all KIMRA measurements (blue X), and coincident MLS measurements (magenta square). The same is shown for all MIRA 2 measurements (green X) and coincident MLS measurements (magenta triangle). The black lines indicate the edges of the polar vortex (see Figure 6).**

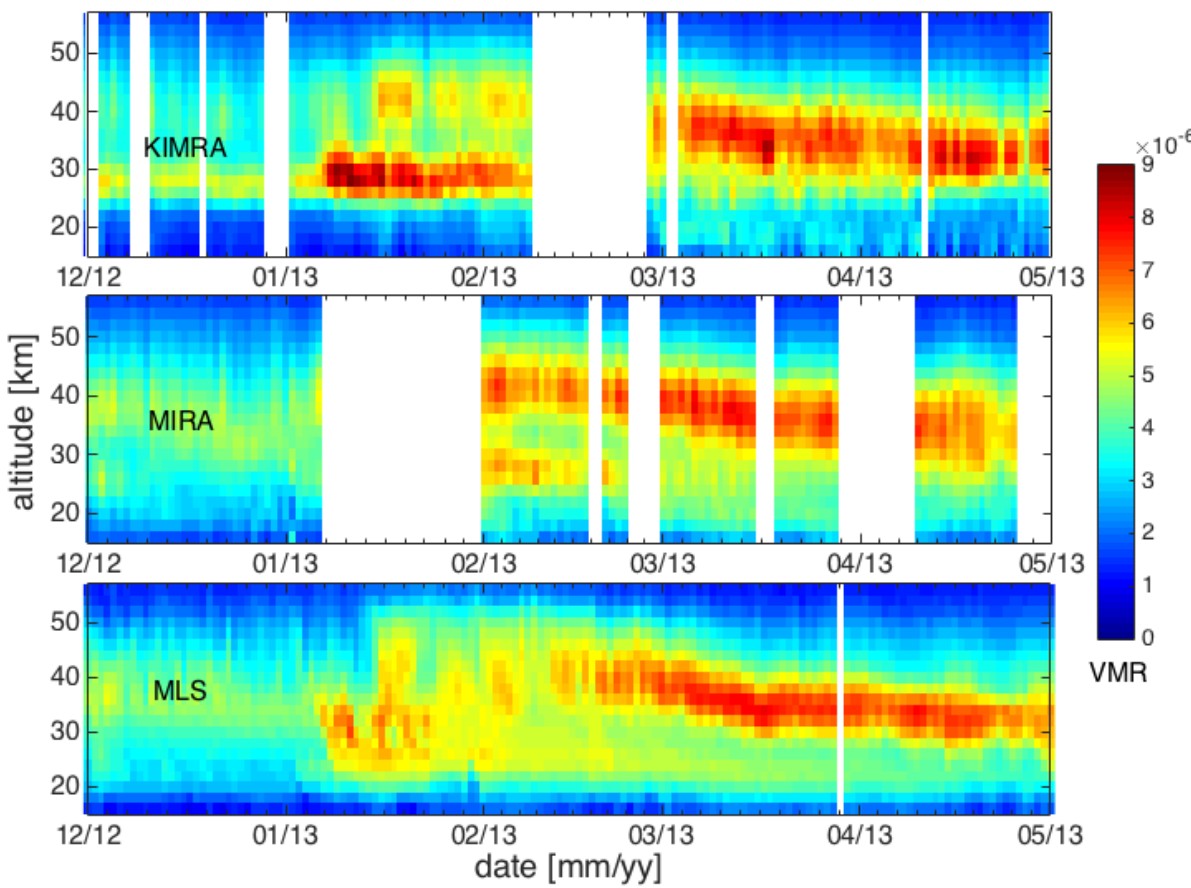

**Figure 12: Time series of the daily average O$_3$ profiles (in ppv) for KIMRA, MIRA 2, and MLS measurements.**







**Figure 13:** Retrieved O$_3$ concentrations (in ppv) above Kiruna from KIMRA, for January to March for the years 2008, 2009, 2010, 2011, and 2013.





**Figure 14: Left: monthly averages of daily Kiruna O₃ profiles (in ppv) from KIMRA measurements for January, February, and March by year. A missing year means that there were not enough measurements during the month to produce a representative average. Right: mean and standard deviations of the monthly profiles.**