# Peer review of "Ozone profiles above Kiruna from two ground-based radiometers"

_Atmospheric Measurement Techniques, 2016_

## Referee Comment (RC1) · Anonymous Referee #1 · 11 May 2016

This is a nice validation paper which focuses primarily on the 2013 period when MIRA and KIMRA were both deployed at Kiruna. The validation information is encompassed primarily in two sets of figures. Figures 3, 7, and 8 show comparisons between average profiles from the instruments, and point-by-point coincident comparisons are presented in Figures 4, 5, 9, and 10. It is therefore important that these figures are as informative as possible in order to provide for the basis for interpretation of the validation.

For Figures 3, 7, and 8, it is unclear what the blue "measurement error" refers to. Is this systematic error? This is unclear in the associated text as well. If the point of these figures plot is to discuss systematic biases, and I think it is, then the appropriate error bars here should be sigma/sqrt(n), not sigma as is shown. Given the number of data points here (e.g. 177 in Figure 3) error bars should then be much smaller. In fact, if the error bars are retained at their current large form, some of the statements made in the

conclusions cannot be drawn. The variability and random error comparison for which the larger error bars would be appropriate is best left to Figures 4, 5, 9, and 10.

For Figures 4, 5, 9, and 10 it is important to clearly define how exactly the x and y errors are determined. To what extent are the slopes sensitive to reasonable variations in the error estimates? A similar study was conducted by Nedoluha et al. [1997], where it was found that different error estimates gave significantly different slopes, but because of the smaller geophysical variations in that study the sensitivity of the slopes to the error estimates may have been much higher. In any case, an estimate of the uncertainty in the slopes based upon an uncertainty in the error estimates should be given. If the slopes are not, within the uncertainty of these estimates, equal to 1, then there is a significant difference in the variations observed by the instruments and this should be discussed. If not, then the appropriate conclusion is that they agree within reasonable uncertainties.

Page 2 - "With a likely upcoming gap in observations from profiling satellite instruments, ground-based instruments will represent the predominant source of atmospheric mea-surements needed to maintain a long-term O3 profile record." While I don't dispute the importance of ground-based instruments, it seems unlikely that there will be a true gap in profiling satellite instruments in the near future. Admittedly MLS may stop operat-ing in the next few years, but OMPS-LP and SAGE III are both likely to be operating for some time, and the OMPS nadir instrument certainly does provide some profile information. Perhaps it would be best to just rephrase this as "With the decrease in ob-servations from profiling satellite instruments, ground-based instruments will represent an increasingly important source . . ."

Page 3 - "as well as two Fast-Fourier-Transform spectrometers (FFTS)." There's only discussion of what is done with the narrowband FFTS. What about the other one?

Page 5 - "Attenuation of the signal due to the troposphere is accounted for by including the Millimeter wave Propagation Model MPM93 H2O continuum (Liebe et al., 1993) in

the inversion." Does this mean that ARTS is not run in the troposphere (i.e. it is run only in the middle atmosphere)? Or does it mean that something is added to ARTS in the tropospheric levels?

Figure 2 - Please put a dashed or thin line at 0.5 ppmv/ppmv (=100% measurement response) to make it easier to estimate the measurement response.

Page 8 - "Either way, the choice of time criterion did not have a substantial effect on the presented results (there was a slight increase in standard deviation)." So there was an increase in standard deviation both for tighter and loser coincidence criteria?

Page 9 – "Both the ozonesonde and MLS profiles were smoothed using the averaging kernels." How were the ozonesonde profiles smoothed with averaging kernels given that their highest altitude is in the middle of the KIMRA/MIRA2 vertical range? It seems surprising that KIMRA shows so much less variation than the sondes in Figure 5, but in other figures that show 16-26km data KIMRA shows more variation the MIRA2 . Any comments on this?

Figure 10: The caption says "same as Figure 10". Presumably it should say "same as Figure 9".

Page 12 – The only reasonable explanation for the double peak structure is the last one given, beginning with "A possible explanation for the observed shape is the combination of downward motion of air within polar vortex, and transport of extra-vortex air into the middle to upper stratosphere". A lot of the discussion leading up to this (chemical ozone depletion, mini-holes, . . .) should be eliminated since it clearly isn't relevant.

Page 12 – "An oscillatory bias was identified in the KIMRA data, present in the comparison with all three instruments." According to Figure 3, 7, and 8 in their current form with their very large error bars, this bias would appear to be insignificant, so it is not clear that this conclusion can be drawn. If the error bars were changed to sigma/sqrt(n) then this conclusion would probably be appropriate.

---

## Referee Comment (RC2) · Anonymous Referee #2 · 19 May 2016

General comments

This is an interesting paper on the validation of the two microwave radiometers KIMRA and MIRA 2 based in Kiruna, Sweden. KIMRA and MIRA 2 are compared to each other when at the same location, to ozone profiles measured by radiosonde (RS) launched from Sodankylä, and to simultaneous measurements by MLS.

The effects on the ozone profiles of a unsolved problem of standing wave in the KIMRA measured spectrum are described. This is leading to a low bias of KIMRA towards MIRA 2, RS and MLS around 22 km.

7 and 5 months mean profiles for the 2012-2013 winter are first compared, then a comparison with RS using regression is performed with the distinction between measurement inside and outside the vortex. Finally, a 5 winters climatology of KIMRA is

used to assess the presence of a dip in the arctic winter ozone profile at 35 km. A qualitative explanation for the presence of this dip in the ozone profile is given and the necessity of further investigations is mentioned.

The paper is clear and well written with good quality figures. The scientific contribution is relevant for publication and lies within the scope of AMT. The methods used for the comparisons are valid, and the related work is referenced. The paper will make a good contribution to AMT, provided that the following comments are addressed.

Specific comments

P1, line16: "KIMRA is low-biased with respect to the ozonesonde data due to a general low bias in the KIMRA profiles around 22 km altitude," A low bias due to a general low bias looks redundant. Please, modify in order to make clear that KIMRA is low biased with respect to radiosonde, MIRA 2 and MLS.

P2, line25-26: To what extent is the inversion procedure presented here different from the older one? Were the older KIMRA spectra showing a similar standing wave? Was the older inversion setup able to deal with that? Please, describe shortly the changes with respect to the previous retrieval setup.

P3, line12: The authors mentioned the two FFTS of KIMRA but only the characteristics of the narrowband FFT are mentioned. What are the characteristics of the second FFTS? Please add.

P3, line 13: "Narrowband often centered": please, mention that the FFTS can be moved to another frequency here instead of later in the text at p3, line 29-33.

P5, line 8: Is the uncertainty estimation of 1 ppmv constant for the whole altitude range? The standing wave on the wings of the spectrum should affect only the bottom of the profile? Please, describe the variation of the 1 ppmv uncertainty with altitude.

P5, line12: Oscillations in the baseline are due to reflections along the quasi-optic path. The distance of the reflection to the horn can be deduced from the oscillation

frequency. Were the authors able to determine in which of the components of KIMRA the reflections are set?

P5, line 22 and Figure 2: The measurement contribution (MC) is 140% at 45 km and 120% at 18km. Can such deviations from 100% be explained? Please, explain the high MC values. As the MC is the sum of the surfaces of the AVK, the shape of the envelope of the AVK should correspond to the shape of the MC profile. This is not the case in Figure 2. Please comment.

P6, line15 and Figure 3 Measurement error of KIMRA resp. Mira 2: the whiskers are the 1 standard deviation of the differences. Are the dashed blue lines, the observation errors which are related to the measurement covariance matrix? In that case, the errors should be minimum in the middle part of the profile where the SNR is maximum? What is exactly the dashed blue line? Please modify in order to clarify what the "the sum of the average measurement error" is.

Does considering the standard deviation/sqrt(n) instead of the standard deviation of the n differences change the conclusions of section 4.2? Same comments for Figure 7 and 8 and conclusions of section 5.3. Please comment.

P9, line 22-23 and Figure 5: Is the number of coincidences influencing the regression coefficient? The statement of higher correlation for MIRA 2 and RS is done on 25 co-incidences for KIMRA vs RS and 13 coincidences for MIRA 2 vs RS. Please comment.

The reader cannot deduce from the good r coefficient of MIRA 2 vs RS that the bias ($\pm$ the standard deviation) of the differences between RS and radiometers is within the range of the sum of the measurement errors from RS and MIRA 2. The regression plot and factors without an estimation of the errors are not sufficient to establish the good correspondence between MIRA 2 and RS, please add errors bars to figure 5 or show the profile of the difference.

P12, line 20-21 : The authors emphasized that the arctic winter dip in ozone at 35 km

is not a result of the biases in KIMRA ozone profiles, but an issue could be : to what extent the bias in KIMRA ozone profiles, bias related to the presence of the standing wave in the measured spectra, is enhancing the ozone dip at 35 km or the maximum intensity at 27 km?

Do the authors have any suggestions? Is it possible to correlate the intensity of the ozone dip with the opacity of the troposphere or with the intensity of the standing wave? How is the standing wave in winter 2008, when the ozone dip is not as clear?

MLS show the ozone dip in Figure 12. Are the MLS profiles AVK smoothed by KIMRA? What is the influence of the smoothing by KIMRA AVK on the ozone dip intensity measured by MLS? Please describe the eventual effects of AVK smoothing of MLS ozone profiles by KIMRA AVK on the ozone dip measured by MLS.

Technical comments

Figure 4, righthand side: up left panel: in the text p6, line 29: slope=0.81, in the figure: slope=0.9; down left panel: in the text p6, line 32: slope=1.0, in the figure, slope=1.07. Please make it consistent.

P8, line 10: Livesey (2008) is not in the reference list

P10, line9: "…shows better agreement with MLS." Please, mention here a reference to Figure 8.

P10, line 14: it should be Figure 4 instead of Figure 3

P13, line 27: Calisesi (2003) is not cited in the text

P16, line 6: Palm(2010) should go to P17, line 22

P17, line 17: Nash(1996) is not cited in the text

P19, Figure 2 left and middle: please add a vertical dashed line at MC=100%

P20, Figure 3, legend: a priori "used" for the Inversion

P20, Figure 4, p26 Figure 12, p27 Figure 13: why ppv instead of ppmv? Please adapt for similarity with the others figures.

P25, Figure 10, legend: should be "as Figure 9" instead of "as Figure 10"

---

## Author Comment (AC1) · 19 Jul 2016

General comments:
First, we would like to thank the reviewer for their feedback. We found it useful and believe that the paper has been improved as a result.

In addition, a small bug was found in the code assigning coincidence based on position relative to the vortex. After correcting this code, the number of coincidences for KIMRA and MLS increased, and the specific set of profiles selected has changed somewhat due to KIMRA having measurements in the vortex edge over the measurement time period. There was also a small increase/change in selected profiles for the MIRA 2 and MLS coincidences. No significant difference is seen in the comparison of the profiles, except for the number of measurements, and a small change was found in the column

comparison differences.

A distinction is now made between standing waves (waves that are set up within components of instruments) and baseline waves (wave structures that are present in the baseline of spectra).

Response to Reviewer:

**This is a nice validation paper which focuses primarily on the 2013 period when MIRA and KIMRA were both deployed at Kiruna. The validation information is encompassed primarily in two sets of figures. Figures 3, 7, and 8 show comparisons between average profiles from the instruments, and point-by-point coincident comparisons are presented in Figures 4, 5, 9, and 10. It is therefore important that these figures are as informative as possible in order to provide for the basis for interpretation of the validation.**

**For Figures 3, 7, and 8, it is unclear what the blue "measurement error" refers to. Is this systematic error? This is unclear in the associated text as well. If the point of these figures plot is to discuss systematic biases, and I think it is, then the appropriate error bars here should be sigma/sqrt(n), not sigma as is shown. Given the number of data points here (e.g. 177 in Figure 3) error bars should then be much smaller. In fact, if the error bars are retained at their current large form, some of the statements made in the conclusions cannot be drawn. The variability and random error comparison for which the larger error bars would be appropriate is best left to Figures 4, 5, 9, and 10.**
Measurement error refers to the error due to statistical noise on the spectrum, but this has been removed from the profile comparison figures as there is not a real value in comparing the systematic bias with the measurement error. It was more for a quick comparison of value. The measurement error has been used instead in Figures 4,5,9, and 10, as suggested.

The error bars on the profile comparisons have been changed to include the standard error of the mean, as suggested. The standard deviation of the differences between profiles is also still included as it represents the space in which one instrument's profile is likely to lie in relation to another instrument's.

Text has been added when describing the plots: "The standard error of the mean difference is also shown but is small due to the sample size".

**For Figures 4, 5, 9, and 10 it is important to clearly define how exactly the x and y errors are determined. To what extent are the slopes sensitive to reasonable variations in the error estimates? A similar study was conducted by Nedoluha et al. [1997], where it was found that different error estimates gave significantly different slopes, but because of the smaller geophysical variations in that study the sensitivity of the slopes to the error estimates may have been much higher. In any case, an estimate of the uncertainty in the slopes based upon an uncertainty in the error estimates should be given. If the slopes are not, within the uncertainty of these estimates, equal to 1, then there is a significant difference in the variations observed by the instruments and this should be discussed. If not, then the appropriate conclusion is that they agree within reasonable uncertainties.**
Agreed. Accordingly, some changes have been made to these plots:

First, a mean of the error on the columns in an altitude range was previously being used for all points in that range. This has been changed so that the error for each measurement is properly represented.

The slope is now calculated for two cases. The first case includes only the measurement error (the error due to the statistical noise on the spectrum), which has been estimated as in Section 3.2. The second case includes the sum of 130% measurement error and the mean of the error on the columns. The idea here is to increase the
individual measurement errors to try to account for other errors that might vary statistically, and to include a constant error that does not change over time or depend on the measurement error. The error bars are now included in the plots and the standard error on slopes is also shown, as defined in York et al. (2004). Some discussion of the results is provided in each section as below.

Sec. 4.2: "The regression coefficients (slope and intercept) are calculated for two cases of KIMRA/MIRA 2 partial column error estimates. The first case includes only the measurement error on the profile: the error due to the statistical noise on the spectrum (Rodgers, 1990), to which an offset has been added to account for short scale waves in the spectral baseline (see Section 3.2). The second case is the sum of 130% measurement error and the mean of the measurement errors on each partial column: the former increase is to try to account for other errors that vary statistically (such as errors in the temperature profile), and the latter is to include an error that does not change in magnitude over time or depend on an individual observation. The idea here is that this will help to capture some the variation in the measurements that is neither truly random nor systematic in nature, such as a baseline error. While not based on it, the larger error estimate appears justified when one considers the bias shown in Figure 3. The limits of these slopes and their standard errors define a range that should contain the value of 1 if the measurements agree. A similar approach was used by Nedoluha et al. (1997), in which case the standard deviation of the satellite measurements being compared was added to the errors of the ground-based measurements.

The results for KIMRA and MIRA 2 are plotted in Figure 4, showing the correlations and the slopes and intercepts of the lines of best fit in each case. The correlations between the partial columns are high, even for layers containing the altitudes with poor profile correlation (Figure 3 (right)), with 36 - 46 km having the highest value of 0.97. A value of 1 lies within the range of the slopes calculated for the two lowermost columns, albeit just barely for the 16 – 26 km column. The 26 – 36 km columns agree for both cases of error estimation. A value of 1 does not lie in the range of slopes for the two

higher partial columns, with MIRA 2 showing a larger range of $O_3$ values in both cases. A value of 1 for the slope does lie in twice the standard error range of the higher error estimates, but for the 46 – 56 km partial column this error is likely an overestimation as the error bars are larger than the variation of the points from the line of best fit."

Sec. 5.3: "Lines of best fit were calculated accounting for errors in X and Y. The correlations between KIMRA and MLS (Figure 9 (lower)) vary between 0.66 and 0.80, and slopes of best fit for the partial columns vary between 0.81 and 0.96, for the case of the lower error estimate. Only for the lowermost column does a value of 1 lie in twice the standard error range of the calculated slopes but it should be noted that the slopes for the lower error estimate all lie within 19% of 1. The correlations between MIRA 2 and MLS (Figure 10 (lower)) are high, between 0.88 and 0.94, and a value of 1 lies in the range of calculated slopes for the two lowermost columns. It can be seen from the two 46 – 56 km panels in Figure 10 that MIRA 2 is low-biased in the case of high $O_3$ columns at these altitudes.

In most instances for the comparisons with MLS, the higher error estimate has a small change (< 0.03) on the value of the calculated slopes, but a large change is seen for the two highest columns in the KIMRA and MLS comparison in Figure 9 (lower). This is likely due to the smaller natural variation in $O_3$ at these altitudes and the presence of outliers in the KIMRA data. MIRA 2 in general shows better agreement with MLS, compared to KIMRA."

**Page 2 - "With a likely upcoming gap in observations from profiling satellite instruments, ground-based instruments will represent the predominant source of atmospheric measurements needed to maintain a long-term O3 profile record." While I don't dispute the importance of ground-based instruments, it seems unlikely that there will be a true gap in profiling satellite instruments in the near future. Admittedly MLS may stop operating in the next few years, but OMPS-LP and SAGE III are both likely to be operating for some time, and the OMPS nadir**

**instrument certainly does provide some profile information. Perhaps it would be best to just rephrase this as "With the decrease in observations from profiling satellite instruments, ground-based instruments will represent an increasingly important source . . ."**

Yes, thank you. This line has been edited to state: "With the decrease in observations from profiling satellite instruments, ground-based instruments will represent an increasingly important source of measurements needed to maintain a long-term stratospheric $O_3$ profile record."

**Page 3 - "as well as two Fast-Fourier-Transform spectrometers (FFTS)." There's only discussion of what is done with the narrowband FFTS. What about the other one?**

The text has been modified to: "The narrowband FFTS, installed in 2007, is often centered on a nearby CO line and has been used in retrieving CO between 40 and 80 km (Hoffmann et al., 2011), and the broadband FFTS, installed in 2012, has been used to measure atmospheric spectra in the region of 230 GHz. The data from the AOS is presented here as it extends back to 2002 and the spectrometer is the same model as the MIRA 2 spectrometer."

**Page 5 - "Attenuation of the signal due to the troposphere is accounted for by including the Millimeter wave Propagation Model MPM93 H2O continuum (Liebe et al., 1993) in the inversion." Does this mean that ARTS is not run in the troposphere (i.e. it is run only in the middle atmosphere)? Or does it mean that something is added to ARTS in the tropospheric levels?**

ARTS is run in the troposphere as well as the stratosphere and the water vapour continuum is included in the model. In an effort to be clearer, this line has been changed to: "Attenuation of the signal due to water vapour, mainly in the troposphere, is accounted for with the Millimeter wave Propagation Model MPM93 $H_2O$ continuum

(Liebe et al., 1993), which can be included in the forward modelling with ARTS."

**Figure 2 - Please put a dashed or thin line at 0.5 ppmv/ppmv (=100% measurement response) to make it easier to estimate the measurement response.**
To better show the measurement response cutoff, a line at 0.4 ppmv/ppmv has been included and the shading removed. And a line of text has been added stating: "The mean measurement response for KIMRA dips just below 0.8 at 35 km due to some negative values in the corresponding averaging kernel but the inversion is still defined as useable here."

**Page 8 - "Either way, the choice of time criterion did not have a substantial effect on the presented results (there was a slight increase in standard deviation)." So there was an increase in standard deviation both for tighter and loser coincidence criteria?**
It was for a looser criterion. The text has been modified to read: "Either way, the choice of time criterion did not have a substantial effect on the presented comparison results (there was a slight increase in standard deviation of the differences for a looser time coincidence)."

**Page 9 – "Both the ozonesonde and MLS profiles were smoothed using the averaging kernels." How were the ozonesonde profiles smoothed with averaging kernels given that their highest altitude is in the middle of the KIMRA/MIRA2 vertical range?**
Good point. The ozonesonde profiles were extended using a scaled KIMRA/MIRA2 a priori profile. This information is now included in this section. "Because the ground-based measurements have some sensitivity to $O_3$ at altitudes higher than the reach of the sondes, the sonde profiles were extended above their maximum altitudes prior to performing the smoothing calculation. This was done using the a priori concentrations

(Section 3.1) scaled to match the sonde data at its highest altitude."

**It seems surprising that KIMRA shows so much less variation than the sondes in Figure 5, but in other figures that show 16-26km data KIMRA shows more variation the MIRA2 . Any comments on this?**
Both KIMRA and MIRA show less variation compared to the sondes than to MLS, but particularly KIMRA, and the reason is unclear. Identification of a cause falls outside the scope of this work and we would not like to make a conjecture.

**Figure 10: The caption says "same as Figure 10". Presumably it should say "same as Figure 9".**
The text has been changed.

**Page 12 – The only reasonable explanation for the double peak structure is the last one given, beginning with "A possible explanation for the observed shape is the combination of downward motion of air within polar vortex, and transport of extra-vortex air into the middle to upper stratosphere". A lot of the discussion leading up to this (chemical ozone depletion, mini-holes, . . .) should be eliminated since it clearly isn't relevant.**
The text has been modified to exclude some of the information.

All information explaining the ozone mini-holes has been deleted. Only reduced information about the measurements in 2002/2003 is retained because it concerns the observation of a structure in the ozone profile that is similar, and therefore relevant, to the structure seen in this work. Also, discussion of chemical ozone depletion and variation of the vortex edge has been retained as these are possible causes that need to be eliminated through this discussion.

Sec. 6.1: "The $O_3$ dip is present for some period of time in each year and disappears

in late February or March. It is persistent up to the end of March in 2009. It is very unlikely that this feature is caused by chemical ozone depletion as ozone loss resulting from heterogeneous reactions in the lower stratosphere has never been seen extending to this altitude in the Arctic (e.g., Manney et al., 2003, 2015; Kuttippurath et al., 2010; Livesey et al., 2015). A strong $O_3$ dip (most similar to 2010 presented here) has been observed previously with KIMRA in the winter of 2002/2003 (Raffalski et al., 2005). This coincided with ozone mini-holes between 4 and 11 December 2002, as reported by the European Ozone Research Coordinating Unit (EORCU), but the KIMRA measurements presented for that winter still show the structure of an $O_3$ dip throughout most of December. The latitudinal extent of the polar vortex has been shown to vary with altitude (e.g., Schoeberl et al., 1992; Manney et al., 1995; Harvey et al., 2002), which could explain an occurrence of a local minimum/maximum, but such a feature would not remain stable long enough to account for the observations shown here."

**Page 12 – "An oscillatory bias was identified in the KIMRA data, present in the comparison with all three instruments." According to Figure 3, 7, and 8 in their current form with their very large error bars, this bias would appear to be insignificant, so it is not clear that this conclusion can be drawn. If the error bars were changed to sigma/sqrt(n) then this conclusion would probably be appropriate.**
Figures 3, 7, and 8 have been modified to show the standard error of the mean.

The text in this section has been edited to: "An oscillatory bias was identified in the KIMRA data: There is a low bias of 1 ppm at 22 km, and a high bias of 1 ppm at 28 km, both with a halfwidth of 5 km."

---

## Author Comment (AC2) · 19 Jul 2016

General comments:
First, we would like to thank the reviewer for their feedback. We found it useful and believe that the paper has been improved as a result.

In addition, a small bug was found in the code assigning coincidence based on position relative to the vortex. After correcting this code, the number of coincidences for KIMRA and MLS increased, and the specific set of profiles selected has changed somewhat due to KIMRA having measurements in the vortex edge over the measurement time period. There was also a small increase/change in selected profiles for the MIRA 2 and MLS coincidences. No significant difference is seen in the comparison of the profiles, except for the number of measurements, and a small change was found in the column

comparison differences.

A distinction is now made between standing waves (waves that are set up within components of instruments) and baseline waves (wave structures that are present in the baseline of spectra).

Response to Reviewer:

**General comments**

**This is an interesting paper on the validation of the two microwave radiometers KIMRA and MIRA 2 based in Kiruna, Sweden. KIMRA and MIRA 2 are compared to each other when at the same location, to ozone profiles measured by radiosonde (RS) launched from Sodankylä, and to simultaneous measurements by MLS.**

**The effects on the ozone profiles of a unsolved problem of standing wave in the KIMRA measured spectrum are described. This is leading to a low bias of KIMRA towards MIRA 2, RS and MLS around 22 km.**

**7 and 5 months mean profiles for the 2012-2013 winter are first compared, then a comparison with RS using regression is performed with the distinction between measurement inside and outside the vortex. Finally, a 5 winters climatology of KIMRA is used to assess the presence of a dip in the arctic winter ozone profile at 35 km. A qualitative explanation for the presence of this dip in the ozone profile is given and the necessity of further investigations is mentioned.**

**The paper is clear and well written with good quality figures. The scientific contribution is relevant for publication and lies within the scope of AMT. The methods used for the comparisons are valid, and the related work is referenced. The paper will make a good contribution to AMT, provided that the following comments are addressed.**

**P1, line16: "KIMRA is low-biased with respect to the ozonesonde data due to a**

**general low bias in the KIMRA profiles around 22 km altitude," A low bias due to a general low bias looks redundant. Please, modify in order to make clear that KIMRA is low biased with respect to radiosonde, MIRA 2 and MLS.**

Agreed, thank you. The abstract has been edited and now states: "KIMRA has a correlation of 0.82 but shows a low bias with respect to the ozonesonde data, and MIRA 2 shows a smaller magnitude low bias and a 0.98 correlation coefficient. Both radiometers are in general agreement with each other and with MLS data, showing high correlation coefficients, but there are differences between measurements that are not explained by random errors."

**P2, line25-26: To what extent is the inversion procedure presented here different from the older one? Were the older KIMRA spectra showing a similar standing wave? Was the older inversion setup able to deal with that? Please, describe shortly the changes with respect to the previous retrieval setup.**

The goal for this work was to develop a retrieval scheme that could be applied as consistently as possible to both instruments. The previous retrieval methods for KIMRA and MIRA 2 employed variations on the OEM and are described in the references given in the text. Also, a different ozone transition has been used for the KIMRA retrievals. As noted in Raffalski et al. (2005), there were standing waves present in older spectra measured by KIMRA and a careful selection of the spectral region was also used to minimize the impacts of these standing waves.

**P3, line12: The authors mentioned the two FFTS of KIMRA but only the characteristics of the narrowband FFT are mentioned. What are the characteristics of the second FFTS? Please add.**

As noted in the response to Reviewer 1, information has been added to the section, reading: "The narrow-band FFTS, installed in 2007, is often centered on a nearby CO line and has been used in retrieving CO between 40 and 80 km (Hoffmann et

al., 2011), and the broadband FFTS, installed in 2012, has been used to measure atmospheric spectra in the region of 230 GHz. The data from the AOS are presented here as its time series extends back to 2002 and the spectrometer is the same model as the MIRA 2 spectrometer."

**P3, line 13: "Narrowband often centered": please, mention that the FFTS can be moved to another frequency here instead of later in the text at p3, line 29-33.**
This has been clarified in the text with the following addition. "KIMRA operates in the frequency range between 195 GHz and 233 GHz. The instrument has the capability to measure many species by tuning within this frequency range but, due to baseline issues, has only been used to measure $O_3$ and, since 2007, carbon monoxide (CO)."

**P5, line 8: Is the uncertainty estimation of 1 ppmv constant for the whole altitude range? The standing wave on the wings of the spectrum should affect only the bottom of the profile? Please, describe the variation of the 1 ppmv uncertainty with altitude.**
The estimates of the errors were made using sensitivity tests and so do not provide information about the altitude structure of "real" baseline waves in the spectra. The text has been modified to clarify this: "These error estimates are based on results of sensitivity tests and do not provide information about the vertical structure of errors caused by "actual" baseline waves in the spectra."

**P5, line12: Oscillations in the baseline are due to reflections along the quasi-optic path. The distance of the reflection to the horn can be deduced from the oscillation frequency. Were the authors able to determine in which of the components of KIMRA the reflections are set?**
This has been attempted but the origin is still unclear for KIMRA. Oscillations in the baseline do not only come from reflections in the quasioptical path, but can also be

due, for example, to impedance mismatching in any part of the system. In looking at the KIMRA spectra, there appears to be a combination of more than one signal.

A recent servicing of the AOS has eliminated the visible oscillation in the MIRA 2 spectra. This was likely an artifact produced by a laser diode at the end of its life.

This information has been included in Section 3.2 and 7.

Sec. 3.2: "The clearly visible structure in the baseline of the MIRA 2 spectra (seen in Figure 1) has been eliminated during recent servicing of the AOS. This structure was likely an artifact produced by a laser diode at the end of its life."

Sec. 7: "A recent servicing of the MIRA 2 AOS has eliminated the visible oscillation in the MIRA 2 spectra."

**P5, line 22 and Figure 2: The measurement contribution (MC) is 140% at 45 km and 120% at 18km. Can such deviations from 100% be explained? Please, explain the high MC values.**
Higher than 100% measurement response is due to the averaging kernel for that altitude having an area greater than 1. The concentration at an altitude is not only sensitive to variations at that altitude, but also to concentrations at other altitudes because of the vertical resolution of the retrieved profile being lower than the retrieval grid. This is often seen for ground based instruments (e.g. Palm et al., 2010, Hoffmann et al., 2011) and to some extent for satellites (e.g. MLS data quality documents – e.g. Livesey et al., 2013).

The values are relatively high here and are likely because the altitude resolution of the retrieved profile at those altitudes is coarser than the altitude resolution of the retrieval grid.

**As the MC is the sum of the surfaces of the AVK, the shape of the envelope of**

**the AVK should correspond to the shape of the MC profile. This is not the case in Figure 2. Please comment.**
The shape of the measurement response generally does not follow the envelope of the averaging kernels. To make an example, one very broad averaging kernel may follow the envelope but have a relatively high area. This is also true for satellites and can be seen in the ozone averaging kernels for MLS (MLS data quality document v4.2; Livesey et al., Version 4.2x Level 2 data quality and description document, Tech. Rep. JPL D-33509 Rev. B, Jet Propulsion Laboratory, Pasadena, Calif., 2016).

**P6, line15 and Figure 3 Measurement error of KIMRA resp. Mira 2: the whiskers are the 1 standard deviation of the differences. Are the dashed blue lines, the observation errors which are related to the measurement covariance matrix? In that case, the errors should be minimum in the middle part of the profile where the SNR is maximum? What is exactly the dashed blue line? Please modify in order to clarify what the "the sum of the average measurement error" is.**
Measurement error does refer to error from the measurement noise covariance matrix. This has been clarified in the text: "The regression coefficients (slope and intercept) are calculated for two cases of KIMRA/MIRA 2 partial column error estimates. The first case includes only the measurement error on the profile: the error due to the statistical noise on the spectrum (Rodgers, 1990), to which an offset has been added to account for short scale waves in the spectral baseline (see Section 3.2)."

The minimum in the relative error for KIMRA and MIRA 2 does occur in this region. This minimum in the absolute (ppmv) error does not, in general, have to occur where the concentration of the trace gas is a maximum. An example is the CO profile which increases strongly with altitude in the mesosphere and has higher absolute errors at these altitude (Hoffmann et al. 2011).

As noted in the response to Reviewer 1, the measurement error has been removed from the profile comparison figures as there is no real value in comparing the systematic bias with the measurement error. It was more for a quick comparison of value. This error has been used in Figures 4,5,9, and 10 instead.

**Does considering the standard deviation/sqrt(n) instead of the standard deviation of the n differences change the conclusions of section 4.2? Same comments for Figure 7 and 8 and conclusions of section 5.3. Please comment.**
Agreed. The error bars on the profile comparisons (Figures 3, 7, and 8) have been changed to include the standard error of the mean, as suggested by both reviewers. The standard deviation of the differences between profiles is also still included as it represents the space in which one instrument's profile is likely to lie in relation to another instrument's.

**P9, line 22-23 and Figure 5: Is the number of coincidences influencing the regression coefficient? The statement of higher correlation for MIRA 2 and RS is done on 25 coincidences for KIMRA vs RS and 13 coincidences for MIRA 2 vs RS. Please comment.**
In theory, the sample size does not affect the regression coefficient. This is not true in practice, in the case of error, and the value obtained will change according to the sample that is taken from a dataset. Increasing the sample size does, however, change the goodness of the fit. The standard errors on the slope are now included with the estimates of the slope and shown on the figures. MIRA 2 shows a larger standard error on the fit than KIMRA does in the case of the sondes.

**The reader cannot deduce from the good r coefficient of MIRA 2 vs RS that the bias ($\pm$ the standard deviation) of the differences between RS and radiometers is within the range of the sum of the measurement errors from RS and MIRA 2. The regression plot and factors without an estimation of the errors are not sufficient to establish the good correspondence between MIRA 2 and RS, please**

**add errors bars to figure 5 or show the profile of the difference.**
Error bars are now included on all column comparison plots and the corresponding text has been modified accordingly. A comparison with measurement error is not made.

**P12, line 20-21 : The authors emphasized that the arctic winter dip in ozone at 35 km is not a result of the biases in KIMRA ozone profiles, but an issue could be : to what extent the bias in KIMRA ozone profiles, bias related to the presence of the standing wave in the measured spectra, is enhancing the ozone dip at 35 km or the maximum intensity at 27 km?**
It is agreed that this point should be emphasized, as well as the fact that the bias in KIMRA can obscure a local minimum in the profile. In comparing to MLS in Figure 12, in early 2013, KIMRA appears to enhance, and in instances create, a minimum in the profile. In late 2013, KIMRA obscures a minimum that is seen in the MLS profile. It will be challenging to accurately separate the two with KIMRA data alone. Text has been modified/added to this effect: "As the location in altitude of this local minimum can change throughout the winter, due to descent of air for instance, the oscillatory bias in the KIMRA profile can either enhance or obscure its presence. Thus, it will be challenging to accurately identify a local minimum feature with KIMRA data alone."

**Do the authors have any suggestions? Is it possible to correlate the intensity of the ozone dip with the opacity of the troposphere or with the intensity of the standing wave? How is the standing wave in winter 2008, when the ozone dip is not as clear?**
A good question. It would be interesting to try to correlate the intensity of the $O_3$ profile at 27/22 km (i.e. the baseline wave signal) with atmospheric opacity. It is possible that that could work to identify a variation in the inversion, and, in the long-run, try to account for some of the bias on an individual profile basis. However, this is speculative and is outside the scope of the work presented in this paper.

It is difficult to quantitatively say what the baseline (or standing wave signal) looked like, but the spectra in 2008 are similar to the spectra in other years.

A reiteration must also be made here that the local minimum is present in the stand-alone MLS data, as shown in Figure 12 for 2013.

**MLS show the ozone dip in Figure 12. Are the MLS profiles AVK smoothed by KIMRA? What is the influence of the smoothing by KIMRA AVK on the ozone dip intensity measured by MLS? Please describe the eventual effects of AVK smoothing of MLS ozone profiles by KIMRA AVK on the ozone dip measured by MLS.**
The MLS data is not smoothed in Figure 12. The caption has been modified to indicate this. The un-smoothed MLS average profile is now included in the profile comparisons in Figure 7 and 8 to show the effect of the smoothing with the averaging kernels of the ground based instruments. The dip is more obviously seen in the average of the MLS profile (non-smoothed included) than in the ground based profiles, but this is only for 2013.

**Technical comments**

**Figure 4, righthand side: up left panel: in the text p6, line 29: slope=0.81, in the figure: slope=0.9; down left panel: in the text p6, line 32: slope=1.0, in the figure, slope=1.07.**
**Please make it consistent.**
Thank you. This has been done.

**P8, line 10: Livesey (2008) is not in the reference list**
This has been added.

**P10, line9: "...shows better agreement with MLS." Please, mention here a reference to Figure 8.**
Done.

**P10, line 14: it should be Figure 4 instead of Figure 3**
This has been changed.

**P13, line 27: Calisesi (2003) is not cited in the text**
It has been removed from the references.

**P16, line 6: Palm(2010) should go to P17, line 22**
Done.

**P17, line 17: Nash(1996) is not cited in the text**
It has been removed from the references.

**P19, Figure 2 left and middle: please add a vertical dashed line at MC=100%**
A vertical line has been added at 80% MC to illustrate the cutoff.

**P20, Figure 3, legend: a priori "used" for the Inversion**
Edited.

**P20, Figure 4, p26 Figure 12, p27 Figure 13: why ppv instead of ppmv? Please adapt for similarity with the others figures.**
They have been changed.

**P25, Figure 10, legend: should be "as Figure 9" instead of "as Figure 10"**
Done. Thank you.